# GeoFormer: Learning Point Cloud Completion with Tri-Plane Integrated Transformer

## ABSTRACT

Point cloud completion aims to recover accurate global geometry and preserve fine-grained local details from partial point clouds. Conventional methods typically predict unseen points directly from 3D point cloud coordinates or use self-projected multi-view depth maps to ease this task. However, these gray-scale depth maps cannot reach multi-view consistency, consequently restricting the performance. In this paper, we introduce a GeoFormer that simultaneously enhances the global geometric structure of the points and improves the local details. Specifically, we design a CCM Feature Enhanced Point Generator to integrate image features from multi-view consistent canonical coordinate maps (CCMs) and align them with pure point features, thereby enhancing the global geometry feature. Additionally, we employ the Multi-scale Geometry-aware Upsampler module to progressively enhance local details. This is achieved through cross attention between the multi-scale features extracted from the partial input and the features derived from previously estimated points. Extensive experiments on the PCN, ShapeNet-55/34, and KITTI benchmarks demonstrate that our GeoFormer outperforms recent methods, achieving the state-of-the-art performance. The code is ready and will be released soon.

## CCS CONCEPTS

• **Computing methodologies** → **Shape inference**; **Point-based models**; **Neural networks**.

## KEYWORDS

Point cloud completion, Canonical coordinate map, Multi-view consistent, Multi-scale Geometry-aware

## 1 INTRODUCTION

Point clouds, arguably the most readily accessible form of data for human perception, understanding, and learning about the 3D world, are typically acquired through ToF cameras, stereo images, and Lidar systems. However, challenges such as self-occlusion, limited depth range of depth camera devices, and sparse output of stereo-matching often result in partial and incomplete point clouds. This presents a significant obstacle for downstream tasks that require a comprehensive understanding of holistic shape. While some object-level point clouds can be obtained through meticulous scanning and fusion techniques, a more efficient approach utilizing deep

Permission to make digital or hard copies of all or part of this work for personal or classroom use is granted without fee provided that copies are not made or distributed for profit or commercial advantage and that copies bear this notice and the full citation on the first page. Copyrights for components of this work owned by others than the author(s) must be honored. Abstracting with credit is permitted. To copy otherwise, or republish, to post on servers or to redistribute to lists, requires prior specific permission and/or a fee. Request permissions from permissions@acm.org.

*ACM MM, 2024, Melbourne, Australia*

© 2024 Copyright held by the owner/author(s). Publication rights licensed to ACM.
ACM ISBN 978-x-xxxx-xxxx-x/YY/MM
https://doi.org/10.1145/nnnnnnn.nnnnnnn

learning has emerged – point cloud completion. This technique is particularly crucial in more challenging scenarios such as robotic simulation and autonomous driving [15, 22, 56].

In recent years, a plethora of deep learning-based methods have emerged [4, 13, 16, 28, 42, 44, 48, 50, 55, 57]. These approaches operate on incomplete 3D point clouds, aiming to predict comprehensive representations. They commonly rely on architectures such as the permutation-invariant PointNet [23] or more advanced transformers [54]. While proficient in global understanding, these permutation-invariant architectures may overly focus on global information and overlook the intrinsic local geometries. Given that point clouds are often sparse and noisy, they struggle to capture geometric semantics accurately, inevitably sacrificing fine-grained details in holistic predictions.

On the contrary, the multi-view projection of point clouds tends to exhibit less noise, as points are aggregated into 2D planes, and semantic information is effectively conveyed through the silhouettes, even incomplete in certain viewpoints. Inspired by the remarkable success of convolutional neural networks (CNN) in the 2D image domain, particularly in tasks super-resolution [6] and inpainting [1], integrating 2D multi-view representations with CNN would hold great promise for 3D point cloud completion. Zhang et al.[53] pioneered the integration of incomplete points with color images as input. However, such an approach necessitates well-calibrated intrinsic parameters, potentially constraining its efficiency and increasing data acquisition costs. In contrast, Zhu et al. [57] utilize multi-view depth maps to enhance data representation and aggregate original input information for high-resolution predictions. Nevertheless, grayscale depth maps offer limited geometric information, thereby constraining the performance of holistic shape prediction, particularly concerning fine-grained details.

To cope with above issues, we propose to incorporate tri-planed projection-based image features with the transformer network structure for point cloud completion, where the three orthogonal planes sufficiently depict the holistic shape. Further, we propose to inject canonical coordinate map (CCM) instead of gray-scale depth map, taking inspiration from recent 3D generation methods [14]. Specifically, we transform point clouds into the canonical coordinate space [30] and treat the coordinates as colors to render image under three orthogonal planes, as shown in Figure 1. CCMs are superior to depth maps for representing point cloud structures and relationships, as multi-view correspondence can be easily reasoning through the color information encoded by CCMs.

However, applying CCMs to point clouds poses a new challenge: objects mapped to canonical space may lose their original scaling. To overcome this, we devise a multi-scale feature augmentation strategy for the partial input point cloud for holistic shape prediction inspired by point upsampling methods [25, 47]. Specifically, we adopt an inception-based 3D feature extraction network from

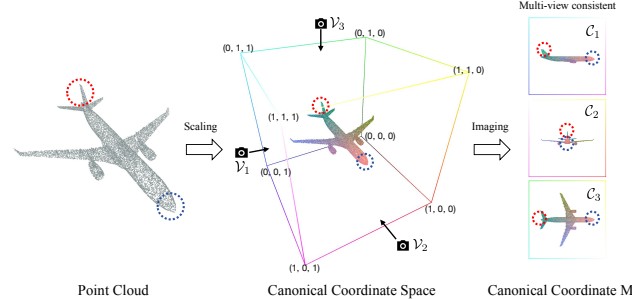

**Figure 1: Illustration of the geometry-consistent tri-plane projection in our GeoFormer. We visualize the details of canonical coordinate maps (CCM) obtained from three orthogonal camera angles and the color of the point represents its normalized coordinate. The highlighted area clearly shows that the three-channel CCM itself contains rich geometric information and ensures multi-view geometric consistency.**

EdgeConv [36] to extract partial input point features. These features, combined with global features using a transformer, predict point offsets. Finally, we integrate these point offsets to obtain the final results, as in previous approaches [55, 57].

In summary, our contributions can be summarized as following:

(1) We introduce multi-view consistent CCMs into point cloud completion, enhancing global features by aligning 3D and 2D features. This is the first work of its kind.

(2) We create an efficient multi-scale geometry-aware upsampler that accurately reconstructs missing parts by incorporating partial geometric features.

(3) We extensively test our method on popular datasets like PCN, ShapeNet-55/34, and KITTI. Results demonstrate our approach has superior performance compared to existing methods, achieving state-of-the-art results across all datasets.

## 2 RELATED WORK

### 2.1 2D Representation Learning of Point Clouds

Point cloud-based representations [23, 24, 36, 54] typically cannot represent topological relations. To address this, [21] introduced a method to establish rich input features, incorporating inductive biases and integrating local as well as global information by projecting 3D point cloud features onto one or multiple 2D planes. However, the point feature is obtained from task-specific neural networks and may lose significant information. In contrast, [53, 57, 58] proposed projecting point clouds into 2D images and utilized a convolution neural network to encode image features directly. However, these 2D features are inconsistent and may destroy the geometric information. Inspired by SweetDreamer [14], which successfully extracted general knowledge from various 3D objects using a diffusion model by learning geometry from the CCM, we attempt to project the partial input point cloud into the canonical coordinate space [30] and obtain multi-view consistent CCMs from three orthogonal views, and design an effective alignment strategy to guide sparse global shape generation and refinement.

## 2.2 Point-based 3D Shape Completion

The point-based completion algorithm is a vital research direction in point cloud completion tasks. These methods [3, 7, 11, 19, 20, 31, 33, 34, 37, 39, 43, 45, 51, 52, 56] usually utilize Multi-layer Perceptions (MLPs) to model each point independently and then obtain global feature through a symmetric function (such as Max-Pooling). Furthermore, voxel-based and transformer-based methods are two important categories of point-based completion approaches.

*2.2.1 Voxel-based Shape Completion.* Early 3D shape completion methods [5, 9, 26] usually rely on voxel grids as 3D object representation. This representation is often applied in a variety of 3D applications [12, 32] because it can be directly and easily processed by 3D convolutional neural network (CNN) architectures. However, to improve performance, these methods inevitably need to increase the voxel resolution which will greatly increase the computational cost. To balance the completion effect and computational overhead, GRNet [44] and VE-PCN [34] choose to utilize voxel grids as intermediate representations and use CNN to predict rough shapes, and then use some refinement strategies to reconstruct detailed results.

*2.2.2 Transformer-based Point Cloud Completion.* Transformer [29] was first proposed for natural language processing tasks and has recently become popular in computer vision areas due to its excellent representation learning capabilities. Recently, this structure was introduced into point cloud completion to extract correlated features between points [4, 13, 16, 17, 31, 41, 42, 48, 49, 54, 55, 57]. These methods can be categorized into two groups according to the upsampling strategy, i.e., point morphing-based methods and coarse-to-fine-based methods. Morphing-based methods [4, 13, 48, 49] first predict point proxies and shape prior features, and then use folding operations proposed by Folding-Net [46] to generate complete point clouds, which usually have a large number of parameters. In contrast, Coarse-to-fine-based methods [16, 17, 41, 42, 50, 55, 57] usually first predict a coarse global structure of point clouds and then utilize some multiple upsampling refinement steps to consider high-quality detail generation. Nevertheless, these methods only exploit limited geometric features and still suffer from robustness and quality issues in accurate completion. Our approach is similar to the coarse-to-fine approaches using the transformer architecture. However, We introduce enhanced global features based on the canonical coordinate map, which can be used for the subsequent coarse prediction and upsampling steps. At the same time, we further design an upsampler that is aware of multi-scale point cloud features to directly predict point coordinates.

## 3 METHOD

### 3.1 Overview

In this section, we will detail our GeoFormer pipeline. Our method mainly consists of one point generator module and two identical upsampler modules, as shown in Figure 2. The point generator module aims to produce sparse yet structurally complete point clouds, and the upsampler module aims to generate complete and dense results from coarse to fine. Specifically, our approach extracts CCM features and aligns them with point cloud features to obtain global geometric representation for coarse point prediction (Section 3.2) and subsequent fine point generation (Section 3.3). Inspired by [16],

Incomplete Point Cloud

Coarse Prediction

Coarse2Fine Generation

Complete Point Cloud

Incomplete Point Cloud & Global Feature $\{\mathcal{P}, \mathcal{F}\}$

**Figure 2: An overview of our pipeline. Given the incomplete point cloud $\mathcal{P}$, we obtain the coarse complete prediction $\mathcal{P}_0$ and extract the global geometric feature $\mathcal{F}$ by utilizing the CCM feature enhanced point generator. In the coarse to fine generation stage, we utilize the multi-scale geometry-aware upsampler to learn coordinate offsets based on $\mathcal{P}, \mathcal{F}$ and previous estimated points $\mathcal{P}_i$, and further scatter them into specific 3D coordinates to reconstruct the accurate and detailed complete result $\mathcal{P}_2$.**

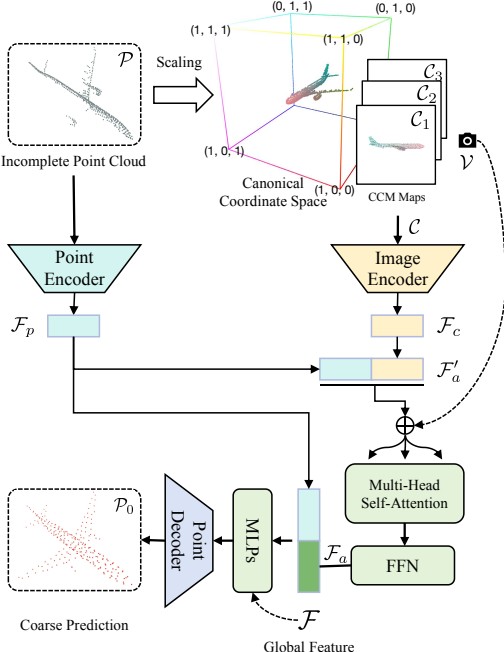

**Figure 3: The detailed structure of the CCM feature enhanced point generator. We first convert partial point cloud input $\mathcal{P}$ into the canonical coordinate space and extract the corresponding projection maps according to the views $\mathcal{V}$. Then, we align the 3D point features and the 2D map features through attention mechanism, and obtain the global features $\mathcal{F}$ after some processing. Finally, we use a 3D coordinate decoder to predict the coarse sparse but complete point cloud $\mathcal{P}_0$.**

we use the chamfer distance loss function in hyperbolic space for constraints (Section 3.4).

## 3.2 CCM Feature Enhanced Point Generator

We propose a novel point generator that aims to produce a sparse yet structurally complete point cloud $\mathcal{P}_0$, and its detailed structure is shown in Figure 3. Analogous to [14, 30], we define the canonical object space as a 3D space contained within a unit cube $\{x, y, z\} \in$

[0, 1]. Specifically, given the partial point cloud $\mathcal{P} \in \mathbb{R}^{N \times 3}$, we first normalize its size by uniformly scaling it so that the maximum extent of its tight bounding box has a length of 1 and starts from the origin. Then, we render coordinate maps $C_i \in \mathbb{R}^{3 \times H \times W}$ from three deterministic views $\mathcal{V}_i \in \mathbb{R}^{3 \times 3}$ for training.

Furthermore, to align the above cross-modalities and predict coarse point clouds effectively, we first use PointNet++[24] to encode $\mathcal{P}$ hierarchically to get $\mathcal{F}_p \in \mathbb{R}^{1 \times 2C}$, and ResNet18[10] as the image encoding backbone network to extract corresponding CCM features $\mathcal{F}_c \in \mathbb{R}^{3 \times C}$ from $C$. To bridge the gap between 2D and 3D features, we propose a novel effective feature alignment strategy. Specifically, we first combine $\mathcal{F}_p$ and $\mathcal{F}_c$ in feature channel-wise to get $\mathcal{F}_a'$ and then use a self-attention architecture with camera pose $\mathcal{V}$ as positional embedding to get fused features $\mathcal{F}_a \in \mathbb{R}^{1 \times 2C}$. Then we can obtain the global geometric semantic feature $\mathcal{F} \in \mathbb{R}^{1 \times 4C}$ by

$$\mathcal{F}_a' = \text{CONCAT}(\mathcal{F}_p, \mathcal{F}_c) \quad (1)$$

$$\mathcal{F}_a = \text{MLP}(\text{MH-SA}(\mathcal{F}_a', \mathcal{V})) \quad (2)$$

$$\mathcal{F} = \text{CONCAT}(\mathcal{F}_p, \mathcal{F}_a) \quad (3)$$

where $\text{CONCAT}(\cdot)$ and $\text{MLP}(\cdot)$ denote channel-wise concatenation operation and multi-layer perception, respectively. $\text{MH-SA}(\cdot)$ denotes the multi-head self-attention transformer, $\mathcal{V}$ is the camera pose embedding. $\mathcal{F}$ aggregates the partial point cloud features and geometric semantic patterns and is employed for subsequent point generation steps.

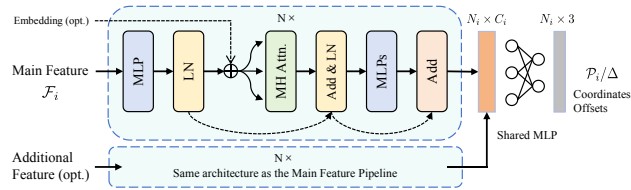

**Figure 4: The detailed structure of the Decoder. We input the main features $\mathcal{F}_i$ into the N networks of attention architecture to get enhanced features, and then we use the shared MLP network to predict 3D coordinates.**

To predict 3D coordinates of coarse complete point clouds $\mathcal{P}_0 \in \mathbb{R}^{N_c \times 3}$, we take transformed $\mathcal{F}$ as input and utilize a decoder to regress the 3D point coordinates directly. What's more, we adopt an operation similar to previous studies [42, 55, 57], where we merge $\mathcal{P}$ and $\mathcal{P}_0$ and resample the merged output for next coarse-to-fine generation steps. The structure of the coordinate decoder is shown in Figure 4, given the previous extracted main feature $\mathcal{F}$, we first transform it to a set of point-wise features using a standard self-attention transformer [29] and then regresses 3D coordinates (point clouds or offsets) with the shared MLPs.

### 3.3 Multi-scale Geometry-aware Upsampler

In the upsampling refinement stage, to reconstruct high-quality details and improve the generalization in real-world point cloud completion, we propose to enhance multi-scale geometric features from partial inputs to guide the upsampling process. Specifically, as shown in Figure 5, we design an inception architecture based EdgeConv [36] to extract multi-scale point features from the partial input $\mathcal{P}$. We use parameters $(i, o, n)$ to define the EdgeConv block, where $i$ is the input channels, $o$ is the output channels, and $n$ is the number of neighbors. We use parameters $(k, o, p)$ to define the 1D Conv block, where $k$ is the kernel size, $o$ is the output channels and $p$ is the padding size. Based on these definitions, we take $\mathcal{P}$ as input and obtain $\mathcal{F}_{e_1} \in \mathbb{R}^{N_p \times C_p}$ and $\mathcal{F}_{e_2} \in \mathbb{R}^{N_p \times C_p'}$ from EdgeConv blocks, which can be defined as:

$$\mathcal{F}_{e_1} = \text{EdgeConv-1}(\mathcal{P}), \mathcal{F}_{e_2} = \text{EdgeConv-2}(\mathcal{F}_{e_1}) \quad (4)$$

where $\text{EdgeConv}(\cdot)$ presents the EdgeConv-based networks with parameters $(i, o, n)$. We further extract multi-scale features $\mathcal{F}_{e_1}' \in \mathbb{R}^{N_p \times 96}$ and $\mathcal{F}_{e_2}' \in \mathbb{R}^{N_p \times 96}$ from previous partial graph-based features with two sets of 1D convolution inception blocks. Then, the final partial input geometry guided features $\mathcal{F}_p'$ can be obtained by:

$$\mathcal{F}_p' = \text{MLP}(\text{CONCAT}(\mathcal{F}_{e_1}', \mathcal{F}_{e_2}')) \quad (5)$$

where

$$\mathcal{F}_{e_1}' = \text{Convs-1}(\mathcal{F}_{e_1}), \mathcal{F}_{e_2}' = \text{Convs-2}(\mathcal{F}_{e_2}) \quad (6)$$

where $\text{Convs}(\cdot)$ defines the inception architecture of multi-scale feature extractor with parameters $(k, o, p)$, $\mathcal{F}_p'$ is transformed through MLPs from $\text{CONCAT}(\mathcal{F}_{e_1}', \mathcal{F}_{e_2}')$ and used to fine points prediction.

To predict fine point clouds, we concatenate features of previous point clouds and $\mathcal{F}$ obtained in previous Section 3.2 to get $\mathcal{F}_{a_i}'$ and use self-attention mechanism to further aggregate these features with additional chamfer distance embedding between partial input $\mathcal{P}$ and previous prediction result $\mathcal{P}_i$ to obtain $\mathcal{F}_{a_i}$ by:

$$\mathcal{F}_{a_i}' = \text{CONCAT}(\text{MLP}(\mathcal{F}), \text{MLP}(\mathcal{P}_i)) \quad (7)$$

$$\mathcal{F}_{a_i} = \text{MH-SA}(\mathcal{F}_{a_i}', \text{CD-}Emb.) \quad (8)$$

Inspired by [57], we take these self-attention features $\mathcal{F}_{a_i}$ as query and $\mathcal{F}_p'$ as key and value to obtain final fused feature $\mathcal{F}_{p_i}$ through cross attention mechanism. Finally, we employ the decoder to predict coordinate offsets $\Delta$ with given ratios to get final refined point clouds $\mathcal{P}_{i+1}$, which can be defined as:

$$\mathcal{F}_{p_i} = \text{MH-CA}(\mathcal{F}_{a_i}, \mathcal{F}_p') \quad (9)$$

$$\Delta = \text{Decoder}(\mathcal{F}_{p_i}, \mathcal{F}_{a_i}) \quad (10)$$

$$\mathcal{P}_{i+1} = \mathcal{P}_i + \Delta \quad (11)$$

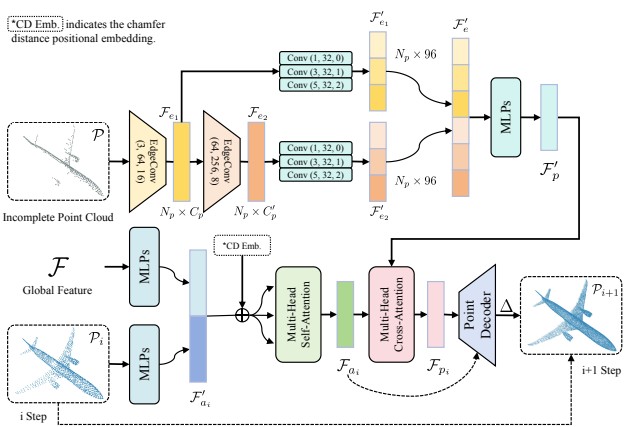

**Figure 5: The detailed structure of the Multi-scale Geometry-aware Upsampler. We design a multi-scale point feature extractor with inception architecture to get local point features $\mathcal{F}_p'$ from partial input $\mathcal{P}$. Then, it is fused with the previous global feature $\mathcal{F}$ and prediction result $\mathcal{P}_i$ to obtain $\mathcal{F}_{p_i}$. Finally, we utilize the decoder to predict the point offset $\Delta$ and obtain the point cloud $\mathcal{P}_{i+1}$. (\*CD Emb. is calculated between $\mathcal{P}$ and $\mathcal{P}_i$)**

where MH-CA$(\cdot)$ denotes the multi-head cross-attention transformer, $\Delta$ is the predicted point offset from Decoder, which is added to the previous result $\mathcal{P}_i$ to get the final result $\mathcal{P}_{i+1}$.

### 3.4 Sensitive-aware Loss Function

To optimize the neural networks, we combine a CD distance loss with a sensitive-aware regularization [16, 18], which helps reduce the negative effects of outliers and improves the generalization ability. Chamfer Distance(CD) measures the differences between the generated point cloud and the ground truth. Given two sets of point clouds $\mathcal{P}$ and $Q$, its general definition is as follows:

$$\text{CD}(\mathcal{P}, Q) = \frac{1}{N} \sum_{p \in \mathcal{P}} \min_{q \in Q} \|p - q\|_2^2 + \frac{1}{M} \sum_{q \in Q} \min_{p \in \mathcal{P}} \|p - q\|_2^2 \quad (12)$$

where $N$ and $M$ represent the number of points in the two sets of point clouds respectively, and $\| \cdot \|_2^2$ represents the Euclidean distance between the points.

However, the classical CD loss function is sensitive to outlier points, limiting point cloud completion performance. Therefore, a recent study [16] proposes to compute CD in hyperbolic space. We further examine the core differences between these CD loss function types, including the general linear function, popular *sqrt* function, and the *arcosh* type loss function proposed by [16]. As shown in Figure 6, the *arcosh*$(1 + x)$ function grows faster near 0, which means it can better distinguish small values and its derivative is always greater than the *sqrt* derivative between $[0, 1]$, which means it can better capture changes in input values. Therefore, *arcosh*$(1+x)$ is more effective as it can avoid local optimal solutions and is anti-overfitting. To summarize, we regularize the training process by computing loss as:

$$\mathcal{L} = \mathcal{L}_{\text{arc-CD}}(\mathcal{P}_0, \mathcal{P}_{gt}) + \sum_{i=1,2} \mathcal{L}_{\text{arc-CD}}(\mathcal{P}_i, \mathcal{P}_{gt}) \quad (13)$$

where

$$\mathcal{L}_{\text{arc-CD}}(x, y) = arcosh(1 + \mathcal{L}_{\text{CD}}(x, y)) \quad (14)$$

## 4 EXPERIMENT

### 4.1 Datasets and Metrics

*4.1.1 Datasets.* We validate and analyze the point cloud completion performance of our proposed GeoFormer on three popular benchmarks, i.e. PCN [50], ShapeNet-55/34 [48] and KITTI [8] Cars dataset, while following the same experimental settings as previous methods [48, 55] (Detailed dataset and implementation details can be found in the supplementary file. Here we only give a brief introduction). **PCN dataset** is one of the most popular benchmarks in point cloud completion, it is a subset of ShapeNet containing shapes from 8 categories. For each shape, this dataset provides 2,048 points as partial inputs and 16,384 points sampled from mesh surfaces as completed ground truth. **ShapeNet-55/34 dataset** is proposed by PoinTr [48], which is also generated from the ShapeNet dataset. However, the ShapeNet-55/34 dataset contains all 55 categories in ShapeNet compared with the PCN dataset, which can test the effect and generalization of the model on a wider variety of objects and unseen categories. At the same time, this dataset provides 8,192 points as ground truth and 3 different difficulty levels test with 2,048, 4,096, and 6,144 points (25%, 50%, and 75% of complete point cloud) which corresponds to simple, moderate, and hard levels. To test our proposed method on real-world scanned objects, we additionally evaluate our method using the **KITTI Cars dataset**, which has 2,401 sparse point cloud objects that are extracted from frames based on the 3D bounding boxes.

*4.1.2 Metrics.* Following [55, 57], we use CD, Density-aware CD (DCD) [40], and F1-Score [27] as evaluation metrics. We report the $\ell^1$ version of CD for the PCN dataset and the $\ell^2$ version of CD for the Shapenet-55/34 dataset. On KITTI Cars benchmark, following the experimental settings of [44, 48, 55], we report two metrics: the Fidelity Distance and Minimal Matching Distance (MMD) performances, which are also developed based on chamfer distance. Detailed definitions can be found in the supplementary.

### 4.2 Comparison with State-of-the-Art Methods

We compare our GeoFormer with many classical methods [28, 35, 44, 46, 50, 52] and several recent state-of-the-art techniques [4, 16, 38, 41, 42, 45, 48, 49, 55, 57]. We conduct extensive experiments on various datasets such as PCN [50], ShapeNet-55/34 [48], and KITTI [8] to demonstrate the effectiveness and generalization of our method.

*4.2.1 Results on the PCN Dataset.* We provide detailed results for each category in Table 1 and compare them with the existing models. We use the best result in their paper for fair comparisons. As shown in the table, our approach outperforms recent methods across all categories, largely improves the quantitative indicators and establishes the new state-of-the-art on this dataset. In Figure 7, we show visual results from three categories (Lamp, Boat, Chair), compared

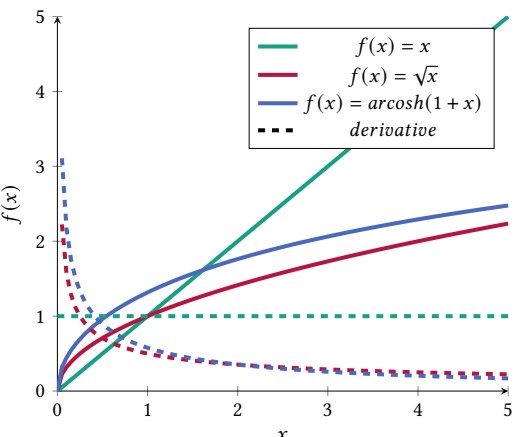

**Figure 6: Illustration of the different chamfer distance post-processing loss functions and their corresponding derivatives**

with PCN [50], GRNet [44], SnowflakeNet [42], PoinTr [48], Seed-Former [55] and SVDFormer [57]. Results show that our method clearly produces superior results with accurate geometry structure and high-quality details. We also provide more visualization results in the supplementary material.

*4.2.2 Results on the ShapeNet-55/34 Dataset.* We further evaluate our method on the ShapeNet-55 benchmark (as shown in Figure 8), which can validate the ability of the model to handle more diverse objects and multiple difficult incompleteness levels. Table 2 reports the overall average $\ell^2$ Chamfer Distance, Density-aware CD and F1-Score results on 55 categories for three different difficulty levels (We show the results for 5 categories with more than 2500 samples in table Table 2. Complete results for all 55 categories are available in the supplementary material). We use CD-S, CD-M and CD-H to represent the CD-$\ell^2$ results under Simple, Moderate, and Hard Settings. Compared with previous methods, our method consistently outperforms, achieving the best scores across all categories and evaluation metrics.

On the ShapeNet-34 benchmark, the networks are challenged to handle novel objects from unseen categories that do not appear in the training phase. We present results on the two test sets at three different difficulty levels in Table 4 (Complete results for all categories are available in the supplementary material). Once again, our proposed approach outperforms others and achieves the best scores, which demonstrates that our method has better performance and generalization ability.

*4.2.3 Results on the KITTI Dataset.* To show the generalization performance of our method in real-world scenarios, we conduct experiments on the KITTI dataset. Following previous methods [33, 44, 55], we finetune our model which pre-trained on the PCN dataset on ShapeNetCars (the cars sub-dataset from ShapeNet [2]) and then evaluate its performance on the KITTI Car dataset for a fair comparison. As shown in Table 3, we report the Fidelity and MMD metrics. Our method obtains better metric scores compared with previous methods.

**Table 1: Quantitative results on the PCN dataset. ($\ell^1$ CD $\times 10^3$ and F-Score@1%)**

| Methods | Plane | Cabinet | Car | Chair | Lamp | Couch | Table | Boat | CD-Avg↓ | DCD-Avg↓ | F1↑ |
|---|---|---|---|---|---|---|---|---|---|---|---|
| FoldingNet [46] | 9.49 | 15.80 | 12.61 | 15.55 | 16.41 | 15.97 | 13.65 | 14.99 | 14.31 | - | - |
| TopNet [28] | 7.61 | 13.31 | 10.90 | 13.82 | 14.44 | 14.78 | 11.22 | 11.12 | 12.15 | - | - |
| PCN [50] | 5.50 | 22.70 | 10.63 | 8.70 | 11.00 | 11.34 | 11.68 | 8.59 | 9.64 | - | 0.695 |
| GRNet [44] | 6.45 | 10.37 | 9.45 | 9.41 | 7.96 | 10.51 | 8.44 | 8.04 | 8.83 | 0.622 | 0.708 |
| CRN [35] | 4.79 | 9.97 | 8.31 | 9.49 | 8.94 | 10.69 | 7.81 | 8.05 | 8.51 | - | 0.652 |
| NSFA [52] | 4.76 | 10.18 | 8.63 | 8.53 | 7.03 | 10.53 | 7.35 | 7.48 | 8.06 | - | 0.734 |
| PoinTr [48] | 4.75 | 10.47 | 8.68 | 9.39 | 7.75 | 10.93 | 7.78 | 7.29 | 8.38 | 0.611 | 0.745 |
| SnowflakeNet [42] | 4.29 | 9.16 | 8.08 | 7.89 | 6.07 | 9.23 | 6.55 | 6.40 | 7.21 | 0.585 | 0.801 |
| PMP-Net++ [38] | 4.39 | 9.96 | 8.53 | 8.09 | 6.06 | 9.82 | 7.17 | 6.52 | 7.56 | 0.611 | 0.781 |
| FBNet [45] | 3.99 | 9.05 | 7.90 | 7.38 | 5.82 | 8.85 | 6.35 | 6.18 | 6.94 | - | - |
| SeedFormer [55] | 3.85 | 9.05 | 8.06 | 7.06 | 5.21 | 8.85 | 6.05 | 5.85 | 6.74 | 0.583 | 0.818 |
| AdaPoinTr [49] | 3.68 | 8.82 | 7.47 | 6.85 | 5.47 | 8.35 | 5.80 | 5.76 | 6.53 | - | 0.845 |
| AnchorFormer [4] | 3.70 | 8.94 | 7.57 | 7.05 | 5.21 | 8.40 | 6.03 | 5.81 | 6.59 | - | - |
| HyperCD [16] | 3.72 | 8.71 | 7.79 | 6.83 | 5.11 | 8.61 | 5.82 | 5.76 | 6.54 | - | - |
| SVDFormer [57] | 3.62 | 8.79 | **7.46** | 6.91 | 5.33 | 8.49 | 5.90 | 5.83 | 6.54 | 0.536 | 0.841 |
| FSC [41] | 4.07 | 9.12 | 8.1 | 7.21 | 5.88 | 9.30 | 6.26 | 6.25 | 7.02 | - | - |
| **Ours** | **3.60** | **8.69** | **7.46** | **6.71** | **5.15** | **8.28** | **5.84** | **5.63** | **6.42** | **0.526** | **0.854** |

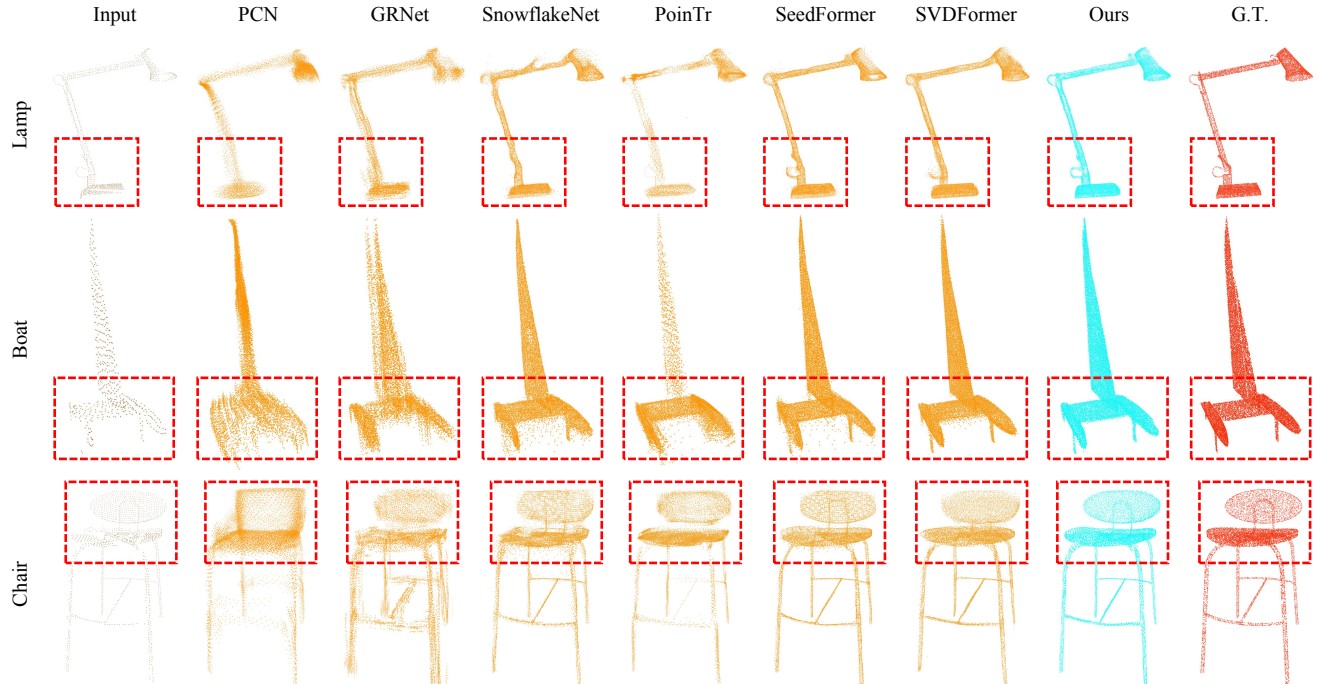

**Figure 7: Visual comparison with recent methods [42, 44, 48, 50, 55, 57] on PCN dataset. Results clearly show that our method can preserve better global structure and reconstruct better local details.**

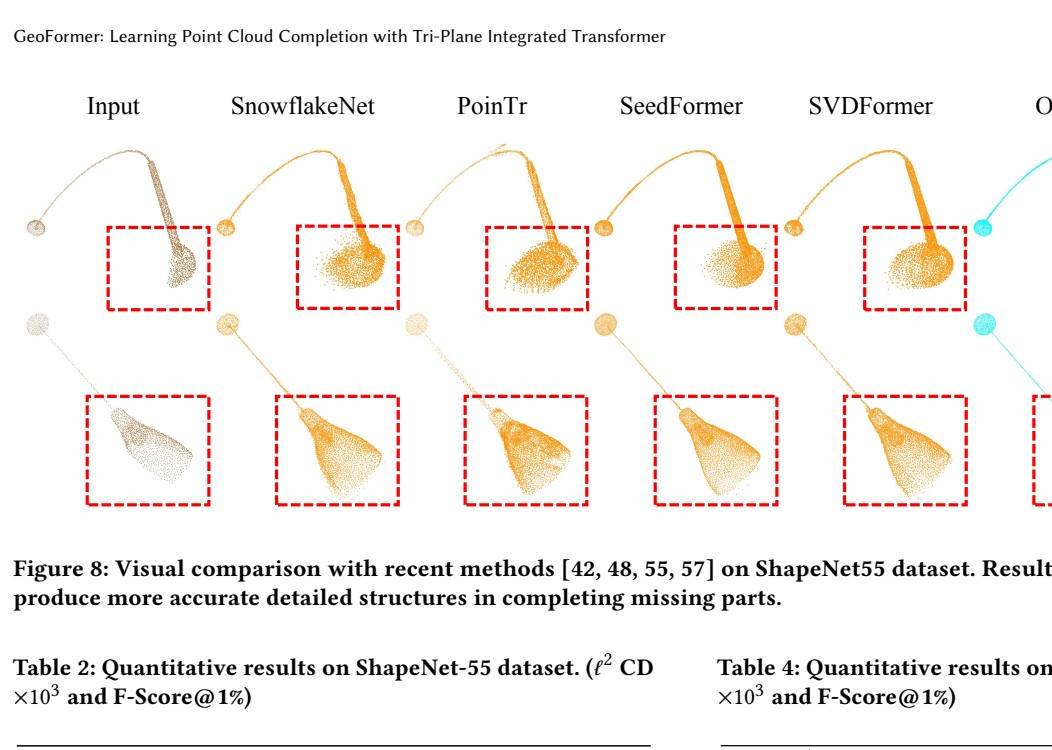

**Figure 8: Visual comparison with recent methods [42, 48, 55, 57] on ShapeNet55 dataset. Results show that our method can produce more accurate detailed structures in completing missing parts.**

**Table 2: Quantitative results on ShapeNet-55 dataset. ($\ell^2$ CD $\times 10^3$ and F-Score@1%)**

| Methods | Table | Chair | Plane | Car | Sofa | CD-S | CD-M | CD-H | CD-Avg↓ | DCD-Avg↓ | F1↑ |
|---|---|---|---|---|---|---|---|---|---|---|---|
| FoldingNet [46] | 2.53 | 2.81 | 1.43 | 1.98 | 2.48 | 2.67 | 2.66 | 4.05 | 3.12 | 0.082 | |
| PCN [50] | 2.13 | 2.29 | 1.02 | 1.85 | 2.06 | 1.94 | 1.96 | 4.08 | 2.66 | 0.618 | 0.133 |
| TopNet [28] | 2.21 | 2.53 | 1.14 | 2.18 | 2.36 | 2.26 | 2.16 | 4.3 | 2.91 | 0.126 | |
| GRNet [44] | 1.63 | 1.88 | 1.02 | 1.64 | 1.72 | 1.35 | 1.71 | 2.85 | 1.97 | 0.592 | 0.238 |
| PoinTr [48] | 0.81 | 0.95 | 0.44 | 0.91 | 0.79 | 0.58 | 0.88 | 1.79 | 1.09 | 0.575 | 0.464 |
| SeedFormer [55] | 0.72 | 0.81 | 0.40 | 0.89 | 0.71 | 0.50 | 0.77 | 1.49 | 0.92 | 0.558 | 0.472 |
| SVDFormer [57] | - | - | - | - | - | 0.48 | 0.70 | 1.30 | 0.83 | 0.541 | 0.451 |
| HyperCD [16] | 0.66 | 0.74 | 0.35 | 0.83 | 0.64 | 0.47 | 0.72 | 1.40 | 0.86 | - | 0.482 |
| **Ours** | **0.58** | **0.65** | **0.34** | **0.69** | **0.57** | **0.41** | **0.64** | **1.25** | **0.77** | **0.540** | **0.514** |

**Table 4: Quantitative results on ShapeNet-34 dataset. ($\ell^2$ CD $\times 10^3$ and F-Score@1%)**

| Methods | 34 seen categories | | | | | | 21 unseen categories | | | | | |
|---|---|---|---|---|---|---|---|---|---|---|---|---|
| | CD-S | CD-M | CD-H | CD-Avg↓ | DCD-Avg↓ | F1↑ | CD-S | CD-M | CD-H | CD-Avg↓ | DCD-Avg↓ | F1↑ |
| FoldingNet [46] | 1.86 | 1.81 | 3.38 | 2.35 | - | 0.139 | 2.76 | 2.74 | 5.36 | 3.62 | - | 0.095 |
| PCN [50] | 1.87 | 1.81 | 2.97 | 2.22 | 0.624 | 0.150 | 3.17 | 3.08 | 5.29 | 3.85 | 0.644 | 0.101 |
| TopNet [50] | 1.77 | 1.61 | 3.54 | 2.31 | - | 0.171 | 2.62 | 2.43 | 5.44 | 3.50 | - | 0.121 |
| GRNet [44] | 1.26 | 1.39 | 2.57 | 1.74 | 0.600 | 0.251 | 1.85 | 2.25 | 4.87 | 2.99 | 0.625 | 0.216 |
| PoinTr [48] | 0.76 | 1.05 | 1.88 | 1.23 | 0.575 | 0.421 | 1.04 | 1.67 | 3.44 | 2.05 | 0.604 | 0.384 |
| SeedFormer [55] | 0.48 | 0.70 | 1.30 | 0.83 | 0.561 | 0.452 | 0.61 | 1.07 | 2.35 | 1.34 | 0.586 | 0.402 |
| HyperCD [16] | 0.46 | 0.67 | 1.24 | 0.79 | - | 0.459 | 0.58 | 1.03 | 2.24 | 1.31 | - | 0.428 |
| SVDFormer [57] | 0.46 | 0.65 | 1.13 | 0.75 | 0.538 | 0.457 | 0.61 | 1.05 | 2.19 | 1.28 | 0.554 | 0.427 |
| **Ours** | **0.39** | **0.57** | **1.05** | **0.67** | **0.537** | **0.515** | **0.55** | **0.99** | **2.15** | **1.23** | **0.551** | **0.483** |

**Table 3: Quantative results on KITTI Cars dataset evaluated as Fidelity Distance and Minimal Matching Distance (MMD) metrics. We follow the previous work to finetune our model on PCNCars.**

| | PCN [50] | FoldingNet [46] | TopNet [28] | GRNet [44] | SeedFormer [55] | **Ours** |
|---|---|---|---|---|---|---|
| Fidelity↓ | 2.235 | 7.467 | 5.354 | 0.816 | 0.151 | **0.089** |
| MMD↓ | 1.366 | 0.537 | 0.636 | 0.568 | 0.516 | **0.510** |

## 4.3 Ablation Studies

In this section, we will demonstrate the effectiveness of the improved design components proposed in our approach. All ablation model variants in the ablation experiments are trained on the PCN dataset with the same settings.

*4.3.1 Loss Function.* The *arcosh* type chamfer distance loss function can effectively reduce over-fitting problems during model training. In the Figure 9 and Table 5, we show the effect of using this loss function alone (variant B, w/o Designs) and the results of adding our proposed improvements (Ours). Results indicate that our designed components can produce more accurate shapes and result in lower CD and DCD scores and higher F1-Score compared to using only the *arcosh* type loss function.

*4.3.2 Our Core Designed Components.* As shown in Figure 10 and Table 6, we compare different ablation variants of our model. Results show that only utilizing the CCM feature as an enhanced semantic pattern (variant C) performs better than the baseline (variant A in Table 5). Furthermore, using only the improved upsampler with a multi-scale inception structure (variant E) introduces geometric priors and shows similar metric improvements as variant C. At the same time, we further add an alignment strategy based on variant C to build the variant D model. The results show that variant D can obtain a lower CD and higher F1-Score. Finally, we combine all designed improved components (Ours) to achieve the best performance across all three metrics.

## 4.4 Complexity analysis

Our method achieves the best performance on almost all metrics on the PCN, ShapeNet-55/34, and KITTI benchmarks. To demonstrate our approach comprehensively and provide a detailed reference for subsequent research, we list the number of model parameters (Params), FLOPs, and inference time (Time) on the PCN dataset of each method in Table 7. All methods are inferred on a single NVIDIA A100 GPU. It can be seen that our method can well balance the computational cost and completion performance.

**Table 5: Effect of loss function and our designs. Results show that arc-CD loss function can improve performance to a certain extent, but our designs are more effective.**

| Variants | arc-CD | Our Designs | CD-Avg↓ | DCD-Avg↓ | F1↑ |
|---|---|---|---|---|---|
| A | | | 6.54 | 0.536 | 0.841 |
| B | ✓ | | 6.50 | 0.535 | 0.846 |
| **Ours** | ✓ | ✓ | **6.42** | **0.526** | **0.854** |

**Table 6: Effect of each parts in our core design components. Results show that both CCM Feature Enhanced Point Generator (Enhance) and Multi-scale Geometry-aware Upsampler (Geometry) can improve the performance individually, and these designs can be combined to get better results.**

| Variants | Enhance | | Geometry | CD-Avg↓ | DCD-Avg↓ | F1↑ |
|---|---|---|---|---|---|---|
| | CCM | Alignment | Inception | | | |
| C | ✓ | | | 6.46 | 0.532 | 0.850 |
| D | ✓ | ✓ | | 6.43 | 0.530 | 0.849 |
| E | | | ✓ | 6.45 | 0.533 | 0.847 |
| **Ours** | ✓ | ✓ | ✓ | **6.42** | **0.526** | **0.854** |

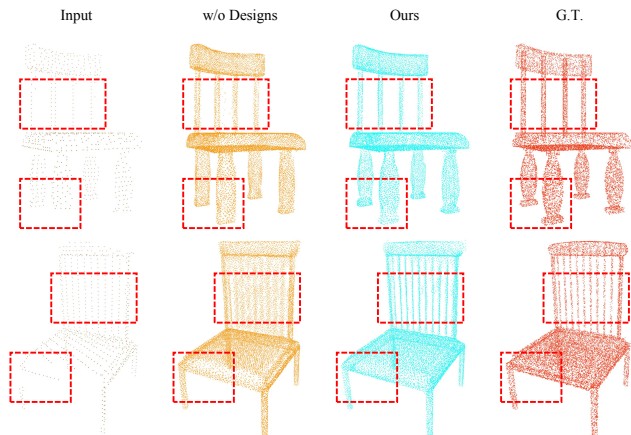

**Figure 9: Visual comparison of variant B (w/o Designs) and Ours (complete approach) on PCN dataset. Results show that using only arc-CD loss without our improved designs may destroy the recovery of fine structures, but our method can reconstruct more accurate details.**

## 5 CONCLUSION

In this paper, we introduce GeoFormer, a novel point cloud completion method aimed at improving completion performance. We propose to extract efficient and multi-view consistent semantic patterns from CCM and then align them with pure point cloud features to enrich the global geometric representation in coarse point prediction stage. Furthermore, we introduce a novel multi-scale feature extractor based on the inception architecture, fostering the generation of high-quality local structure details in point clouds. Our

**Table 7: Complexity analysis. We show the the number of parameter (Params) and FLOPs and inference time (ms) of our method and eight existing methods. We also provide the distance metrics CD-Avg and DCD-Avg on PCN dataset.**

| Methods | Params | FLOPs | Time | CD-Avg↓ | DCD-Avg↓ |
|---|---|---|---|---|---|
| FoldingNet [46] | **2.41M** | 27.65G | - | 14.31 | 0.688 |
| PCN [50] | 6.84M | 14.69G | - | 9.64 | 0.651 |
| GRNet [44] | 76.71M | 25.88G | **10.91ms** | 8.83 | 0.622 |
| PoinTr [48] | 31.28M | 10.60G | 15.35ms | 8.38 | 0.611 |
| SnowflakeNet [42] | 19.32M | 10.32G | 16.65ms | 7.21 | 0.585 |
| SeedFormer [55] | 3.31M | 53.76G | 44.32ms | 6.74 | 0.583 |
| AnchorFormer [4] | 30.46M | **7.27G** | - | 6.59 | - |
| SVDFormer [57] | 32.63M | 39.26G | 18.24ms | 6.54 | 0.536 |
| Ours | 32.76M | 39.37G | 17.41ms | **6.42** | **0.526** |

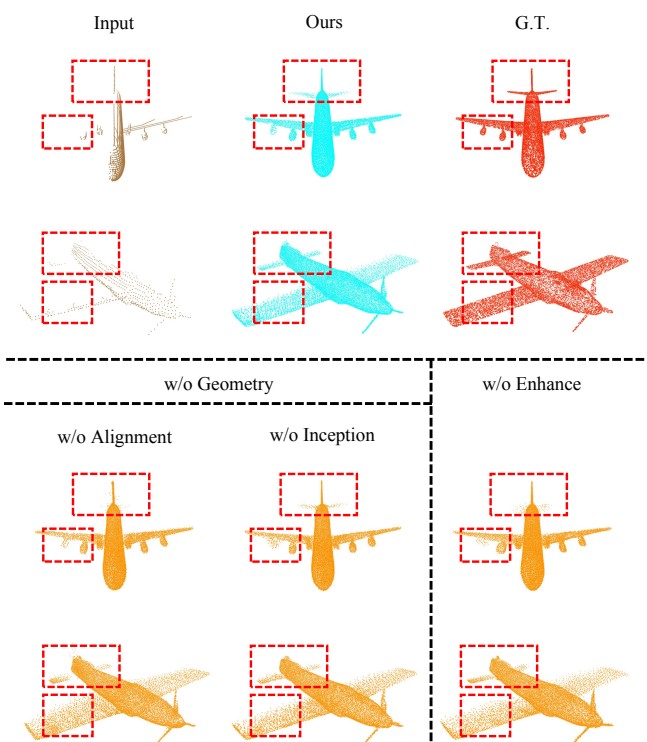

**Figure 10: Visualization comparisons of different design variants. Results show that variant C (w/o Alignment), which only utilizes CCM features, may destroy the global structure. After adding alignment strategy, variant D (w/o Inception) can preserve a better global structure. Variant E (w/o Enhance) only uses the inception structure in upsampling stage and reconstructs dense areas but incomplete shape. In comparison, Ours (complete approach) combines the advantages of these designs and achieves the best results.**

experiments on various benchmark datasets demonstrate the superiority of GeoFormer, as it adeptly captures fine-grained geometry and precisely reconstructs missing parts.

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
