# OpenReview forum: "GeoFormer: Learning Point Cloud Completion with Tri-Plane Integrated Transformer"
_acmmm.org/ACMMM/2024/Conference — MM2024 Poster_

### Official Review · Reviewer_839B · 2024-05-01

**Rating:** 4
**Confidence:** 2

**Summary:**

This paper introduces a method for point cloud completion that aims to recover the accurate global geometry of local point clouds and preserve fine-grained local details by integrating Transformer with Tri-Plane. A point generator for CCM feature enhancement is designed.  In addition, a multi-scale geometric sensing up-sampler module is used to progressively enhance local details through cross-attention between partial inputs and features derived from previously estimated points. Experiments demonstrated that the method outperformed recent methods in PCN, ShapeNet-55/34 and KITTI benchmarks, achieving state-of-the-art performance.

**Strengths:**

1. The introduction of multi-view consistent CCMs enhances global features by combining image features from multi-view consistent canonical coordinate mapping (CCMs) with pure point features, and then aligning 3D and 2D features.
2. An efficient multi-scale geometric sensing upsampler was designed to accurately reconstruct the missing part by combining some geometric features.
3. Extensive testing on popular datasets shows that the method outperforms existing methods on all datasets, achieving state-of-the-art results.

**Limitations:**

1. The work did not mention the computational complexity data of the method and whether the problem is in the running time.
2. Limitations of the proposed approach are not discussed.

**Suitability:**

3

---

### Official Review · Reviewer_i1Wi · 2024-05-20

**Rating:** 5
**Confidence:** 3

**Summary:**

This work introduces a novel GeoFormer approach using a tri-plane integrated transformer for point cloud completion. Firstly, a CCM Feature Enhanced Point Generator was proposed to integrate image features from multi-view consistent canonical coordinate maps (CCMs) and align them with pure point features, thereby enhancing the global geometry feature. Additionally, the Multi-scale Geometry-aware Upsampler module was designed to progressively enhance local details. The proposed method was evaluated on the PCN, ShapeNet-55/34, and KITTI benchmarks for point cloud completion and compared against recent works.

**Strengths:**

1.Clarity and readability: the paper is well written, and the language is very fluent and easy to follow.
2.CCM feature enhanced point generator and multi-scale geometry-aware Upsampler are entirely novel and very innovative.
3.Comparison against recent works.

**Limitations:**

1.Is the point decoder of multi-scale Geometry aware Upsampler in Figure 5 the same as the structure of Figure 4? It doesn't make it clear.
2.The font size of Table 2, Table 3 and Table 4 is too small.
3.For Figure2, why utilize the multi-scale geometry-aware upsampler twice in the coarse to fine generation stage? Whether the number of the multi-scale geometry-aware upsampler has an impact on the performance of the proposed GeoFormer has not been discussed.

**Suitability:**

3

---

### Official Review · Reviewer_KS7A · 2024-05-23

**Rating:** 4
**Confidence:** 3

**Summary:**

This paper introduces multi-view consistent CCMs into point cloud completion, enhancing global features by aligning 3D and 2D features. Extensivel experiments on popular datasets like PCN, ShapeNet-55/34, and KITTI demonstrate that the approach has superior performance compared to existing methods, achieving state-of-the-art results across all datasets.

**Strengths:**

+ The performance on PCN, ShapeNet-55/34, and KITTI are significant.

+ The motivation is good, which uses multiview consistency for completion.

+ The technical pipeline is well described.

**Limitations:**

- My major concerns are the efficacy of this method, as it needs to project the 2D images. Does it need more time for training and inference compared to the current methods using pure point clouds?

- The paper's title contains "Tri-Plane." After reading the paper, I found that tri-planes are three-view projections of the point cloud. In general, tri-planes are usually used to describe the tri-plane for implicitly predicting the SDF values. It is okay to use "Tri-plane", but it causes a little confusion before I finish reading this paper.

**Suitability:**

2

---

### Official Review · Reviewer_orVz · 2024-05-24

**Rating:** 3
**Confidence:** 3

**Summary:**

Like its predecessors seedformer, AnchorFormer, and SVDFormer, this work follows the prevalent transformer-based progressive point cloud completion scheme. The difference is, that it integrates the rendered image information into the global feature extraction for multi-view consistency. Besides, it replaces the CD with a sensitive-aware CD variant as the loss function. Experimental results seem promising.

**Strengths:**

1. Integrating the 2D image information into the completion pipeline.
2. The experiments are thorough and detailed.
3. CD is notorious for its sensitivity to outliers and insensitivity to the distribution difference. This work replaces the CD with a more comprehensive loss function. It also introduces DCD to evaluate the completion results.

**Limitations:**

1. What is the difference between the rendered color image and the canonical coordinate map? The concept of CCM is not well explained in the paper.
2. SVDFormer and this work both utilize 2D and 3D information for point cloud completion and this work achieves an overall superior performance over SVDFormer. Why the CCM feature is a better choice than the depth map is also not clearly illustrated except "Nevertheless, grayscale depth maps offer limited geometric information, thereby constraining the performance of holistic shape
prediction, particularly concerning fine-grained details.". The way I see it, the depth map information is also capable of capturing the holistic shape since both features are obtained from the partial input.
3. This work has its novelty but still resembles its predecessors to a large extent.
4. How exactly are the camera poses embedded into the global feature? Since the CCM maps are fused first and concatenated with the point features, how should the camera pose be used as the positional embedding?
5. Visualized results under real-world settings should also be included. The cases selected from the PCN and ShapeNet dataset are rather easy to complete.

**Suitability:**

3

---

### Meta-Review · Area_Chair_mfmF · 2024-06-28

**Recommendation:** Accept (Poster)
**Confidence:** 4

**Metareview:**

This paper receives mixed scores of weak acceptance and borderline. As it is interesting to use 2D information in point cloud completion task and comprehensive experiments, I recommend a decision of acceptance.

Some reviewers still have concerns on novelty and visualization results. The authors should further clarify them in the final version.